# Arrhythmogenic Cardiomyopathy: Exercise Pitfalls, Role of Connexin-43, and Moving beyond Antiarrhythmics

**DOI:** 10.3390/ijms23158753

**Published:** 2022-08-06

**Authors:** Isabella Leite Coscarella, Maicon Landim-Vieira, José Renato Pinto, Stephen P. Chelko

**Affiliations:** 1Department of Biomedical Sciences, Florida State University College of Medicine, Tallahassee, FL 32303, USA; 2Department of Medicine, Johns Hopkins University School of Medicine, Baltimore, MD 21215, USA

**Keywords:** arrhythmogenic cardiomyopathy, fibrofatty infiltration, connexin-43, exercise

## Abstract

Arrhythmogenic Cardiomyopathy (ACM), a Mendelian disorder that can affect both left and right ventricles, is most often associated with pathogenic desmosomal variants that can lead to fibrofatty replacement of the myocardium, a pathological hallmark of this disease. Current therapies are aimed to prevent the worsening of disease phenotypes and sudden cardiac death (SCD). Despite the use of implantable cardioverter defibrillators (ICDs) there is no present therapy that would mitigate the loss in electrical signal and propagation by these fibrofatty barriers. Recent studies have shown the influence of forced vs. voluntary exercise in a variety of healthy and diseased mice; more specifically, that exercised mice show increased Connexin-43 (Cx43) expression levels. Fascinatingly, increased Cx43 expression ameliorated the abnormal electrical signal conduction in the myocardium of diseased mice. These findings point to a major translational pitfall in current therapeutics for ACM patients, who are advised to completely cease exercising and already demonstrate reduced Cx43 levels at the myocyte intercalated disc. Considering cardiac dysfunction in ACM arises from the loss of cardiomyocytes and electrical signal conduction abnormalities, an increase in Cx43 expression—promoted by low to moderate intensity exercise and/or gene therapy—could very well improve cardiac function in ACM patients.

## 1. Introduction 

Myopathies encompass both inherited and acquired disorders of the muscular system, including congenital myopathies (i.e., present at birth), muscular dystrophies (i.e., progressive skeletal myopathies), endocrine myopathies (e.g., Cushing’s Syndrome), cardiomyopathies (e.g., dilated and hypertrophic), and more recently, the COVID-19 pandemic has resurfaced myocarditis (i.e., infectious myopathies). In our review, we summarize the core clinical and pathological phenotypes of Arrhythmogenic Cardiomyopathy (ACM), an inherited heart disease that can affect both left and right ventricles, or even both, which is plagued by progressive muscle deterioration, fatal arrhythmias, exercise-induced disease penetrance, and cardiac fibrosis. Although ACM is not an infectious cardiomyopathy, it presents with distinct inflammatory pathologies observed in viral and/or bacterial myocarditis. In addition, we evaluate here the ironically cruel influence of exercise on ACM’s pathophysiological progression. A consensus correlation between exercise and increased risk of sudden cardiac death (SCD) restricts patients from physical activities. In truth, patients diagnosed with ACM usually have a complete change of lifestyle in order to prevent disease progression [1].

The proposed mechanism of ACM’s pathogenesis starts with cardiomyocyte-to-cardiomyocyte detachment at intercalated discs (ICDs), a region that promotes myocyte contraction via cell-to-cell adhesion and electrical impulse propagation [2,3]. A distinct pathological feature, myocardial inflammation, supersedes cardiomyocyte loss, and gradual inflammatory infiltration by immune cells in addition to the heart’s innate immune response leads to ACM’s most distinct and ruthless phenotype—progressive fibrofatty replacement of the myocardium [3,4,5,6,7,8,9]. 

Connexin-43 (Cx43) is a gap junction (GJ) protein that has a primary role in cellular communication, coupling, and adhesion. In cardiac tissue, GJ systems are closely associated with the maintenance of electrical impulses by enhancing the intracellular communication between cardiomyocytes [10]. Increased expression of Cx43 ameliorates cardiac dysfunction, however, Cx43 is diminished in ACM patients. In general, Cx43 expression—increased by exercise—may be a missed opportunity to bring both pathological, electrical, and functional benefits in ACM treatment and quality of life (QOL).

The ultimate goal of therapeutics is to provide symptom relief, improved QOL, and mortality prevention—and ACM is no exception. Over the years, physicians and scientists have worked together to thoroughly address the clinicopathologic aspects of ACM and offer effective therapeutics to manage ACM symptoms. Despite the use of both long-standing and modern medicine (e.g., angiotensin-converting enzyme inhibitors [ACEIs], angiotensin receptor blockers [ARBs] amiodarone, sotalol, and beta-blockers) to contribute to health improvement, there are many unstudied—but fundamental—facets of treatment for this life-threatening illness.

This review highlights the commonalities and differences of ACM compared to dilated and hypertrophic cardiomyopathy, evaluates the emergence of refined exercise protocols for translational animal studies, and analyzes “both sides of the Cx43 coin” on physical activity in ACM patients, and the need for precision medicine therapeutics. The latter is of monumental importance, as antiarrhythmics do not repress ACM’s worst phenotype—fibrotic lesions that leave the heart dysfunctional.

## 2. The Cardiomyopathies

Arrhythmogenic Cardiomyopathy, commonly referred to as Arrhythmogenic Right Ventricular Cardiomyopathy (ARVC), is a myocardial disorder that can affect the right (ARVC), the left (ARLV), or both ventricles; hence its recent and more inclusive nomenclature—ACM. Pathologically, the two most uniquely specific traits of ACM are cardiac inflammation and fibrofatty replacement of the myocardium [4]. This progressive myocardial replacement is linked with ICD protein disarrangement that is associated with pathogenic desmosomal variants [11]. These gene variants lead to dysfunctional proteins responsible for structural and electrical connections between cardiomyocytes, leading to cardiomyocyte apoptosis/necrosis. Cardiomyocyte loss is considered one of the triggers leading to myocardial inflammation [12,13], where inflammation precedes fibro-fatty replacement of the myocardium.

The diagnosis of ACM can be elaborate considering symptoms can go undetected for years [14,15]. ACM patients harboring a known pathogenic gene variant may not present with any phenotypes (i.e., asymptomatic gene carrier), even though this identical gene and specific variant may be present in an aged-match symptomatic ACM patient. Hence, ACM is often described as reduced penetrance with variable expressivity. Sadly, and too often, the first presentation is sudden cardiac death/arrest (SCD/A). That said, common features include exertional syncope, cardiac dysfunction (reduced percent ejection fraction [%EF] <40%), >500 ventricular extrasystoles in 24 h, T-wave inversions in precordial leads V1-V6, epsilon waves, ventricular arrhythmias of left bundle branch block (LBBB) pattern, and ventricular tachycardia (VT) [15,16]. Mechanical and electrical dysfunction is more frequently seen in the RV free wall (RVFW) and outflow tract (RVOT), hence ARVC is more commonly used in the field. However, ALVC and biventricular (i.e., ACM) forms have been identified [17]. Magnetic resonance imaging (MRI) can also support diagnosis with findings such as ventricular wall thinning, dyskinesia and hypokinesia, reduced %EF and percent fractional area change (%FAC), and myocardial fibrosis/scar using late gadolinium enhancement-MRI (LGE-MRI) or in some instances, an endomyocardial biopsy [18]. Some cases are difficult to diagnose via MRI and/or echocardiography, thus computed tomography angiography may be recommended. This diagnostic technique can reveal the presence of ventricular wall swelling (i.e., aneurysms) and akinetic regions—once called the “triangle of dysplasia” [19,20]. Considering that fibrofatty tissue can cause electrical conduction block (i.e., re-entrant VT) or persistent arrhythmias even with antiarrhythmics, an electrophysiological study may be necessary to find the site of electrical storms for VT-ablation [21]. Regardless of index presentation, LV dysfunction is a common phenotype at the time of transplantation; where the most frequent attestation for transplant is heart failure (HF) [22]. Although HF was originally considered to be rare in ACM [23], recent studies by Gilotra NA et al. demonstrated that at least one symptom of HF was present in 49% (n = 142/289) of ACM patients [24].

Since there are diagnostic complications, such as patients diagnosed with myocarditis at index presentation, genetic screening can reveal the presence of specific pathogenic variants that are associated with ACM [25]. Considered a “disease of the cardiac desmosome”, as >60% of cases involve pathogenic variants in desmosomal genes, ACM is a familial heart disease with the majority of cases involving autosomal dominant inheritance [18]. Although rare, autosomal recessive inheritance, compound heterozygosity and/or digenic mutations have been documented [18]. The most prevalent mutated protein is in the encoding gene *plakophilin-2* (*PKP2*), critical for intracellular adhesion to the cytoskeleton [26]. Additionally, mutations in desmosomal genes *desmoglein-2* (*DSG2*), *desmocollin-2* (*DSC2*), *desmoplakin* (*DSP*), and junctional *plakoglobin* (*JUP*) are associated with loss of adhesion and altered Ca^2+^ signaling [27,28,29]. Recently identified, mutations in the gene *CDH2*, which encodes a cadherin-2 protein, a major component of the adherent junction system [30], and the ion channel gene *sodium voltage-gated channel alpha subunit-5* (*SCN5A*) are reported to cause ACM through non-canonical pathways [31]. *Desmin* (*DES*) mutations have also been associated with ACM [32]. Correlation analyses of the genetic variants connected to ACM were published by Hoorntje et al., identifying prevalence and inheritance within each mutation [29]. 

In addition to the cardiac phenotypes ACM patients develop, there are, sadly, QOL changes ACM patients will encounter and for some, must implement in everyday life. For example, patients who recently had a VT storm, or an ICD discharge are restricted from driving. Although driving restrictions are dependent upon the state in which the patient resides, driving restrictions can range from 3–12 months. Although we will discuss exercise guidelines below, even a simple day at the beach or lounging at the pool can be a life-altering event. Syncope, VT storm, or an ICD shock may limit a patient’s independence, as a “buddy system” is essential—even at a depth of 2 inches of water—for fear of drowning. Furthermore, ACM patients often confront the burdens of psychological stress, such as ICD implantation, ICD shocks (e.g., appropriate, or inappropriate discharges), and starting a family—as trepidation of passing down a pathogenic variant is possible. Although not all of these QOL changes are applicable to all heart diseases, ACM presents with considerable phenotypic overlap with the other two cardiomyopathies: hypertrophic (HCM) and dilated cardiomyopathy (DCM). 

Dilated Cardiomyopathy is a disease of the cardiac muscle which causes decreased myocardial performance, systolic dysfunction, and ventricular dilation, that may or may not present with hypertension, and can manifest as an ischemic or non-ischemic disease [33,34,35]. Ischemic DCM manifests via extensive myocardial apoptosis/necrosis [36,37] typically following an acute myocardial infarction, while non-ischemic DCM shows no abnormal loading conditions [38] and represents the majority of clinical cases. Around 35% of DCM cases are familial inherited, while the other 65% are acquired throughout the patient’s life. Acquired DCM can arise from environmental infections, autoimmune diseases, endocrine diseases, muscular dystrophy, and/or long-term alcohol abuse [39,40]. DCM symptoms vary and can present as fatigue, weight gain, thromboembolism, and chest pain [41,42]. Although DCM patients present a lower burden of comorbidities, HF—more specifically HF with reduced ejection fraction—is the most common cause of mortality in DCM patients [38]. By the identification of DCM’s features, monitoring and treatment are recommended for patients. Monitoring by electrocardiography (ECG) reports can be somewhat limited, and a variety of ECG morphologies and features are found for both genetic and/or acquired forms of DCM [43]. Some of these findings are LV hypertrophy (LVH) as observed by increased QRS amplitude and duration, left atrial (LA) enlargement seen via a biphasic (“notching”) P-wave, LBBB, atrial fibrillation, repolarization abnormalities, and inferior T-wave inversions [40,44]. Electrical conduction abnormalities are particularly prevalent in inherited DCM, and are likely to present as sinus node dysfunction, sinus node arrest, intraventricular conduction delay, depolarization abnormalities, and supra- and ventricular arrhythmias [45]. Of particular note, the presence of intraventricular conduction dyssynchrony can result in decreased ventricular systolic function and mitral regurgitation, an indication of HF [46,47]. An additional clinical presentation of DCM is elevated levels of troponin-T (TNNT), a serum biomarker indicative of cardiomyocyte degeneration [48]. Myocardial infarction can also be identified through LGE-MRI, which can assess the levels of ischemic myocardium [49]. In inherited DCM, genetic mutations underlying this disease are commonly associated with components of the cytoskeleton and sarcomere [50]. The most common genes encoding sarcomeric proteins that are affected in DCM are: *β-myosin heavy chain* (*MYH7*) [51], *TNNT2* [52], *α-tropomyosin* (*TPM1*) [53], *troponin-C1* (*TNNC1*) [54], *TNNI3* [55], *titin* (*TTN*) [56], and *actin* (*ACTC1*) [57]. Mutations in the structural and functional gene *DES*, a component of the intermediate filament, cause DCM and can be concomitant—or not—with skeletal myopathies [58,59]. Another genetic mutation known to contribute to features of DCM is the nuclear envelope protein *lamin-A/C* (*LMNA/C)* [60], which is also associated with Emery–Dreifuss muscular dystrophy and ACM [61,62,63]. On an important note, different myosin heavy chain isoforms can be found in DCM. The β- and α- isoforms are selectively associated with phenotypic modifications and disease progression to HF [64]. Although a variety of genetic mutations are associated with DCM, the most common findings are reduced Ca^2+^ affinity and contractility [65,66].

Hypertrophic Cardiomyopathy is primarily considered an inherited cardiac disorder and presents distinct characterizations, such as thickening (i.e., hypertrophy) of the LV free wall and septum [67]. The development of LVH is not associated with any other evident cause, such as systemic or metabolic diseases [68], and thus HCM is considered a primary heart disease. Symptoms of HCM vary depending on the severity of the condition and age and include shortness of breath, chest pain, palpitations, fatigue, pulmonary congestion, and swelling of the legs (i.e., peripheral edema) [69,70]. The development of HF in HCM is a major concern due to obstruction of blood flow [71]. Diagnosis of HCM rarely occurs via ECG, as most cases show a normal ECG, yet some distinct features include the presence of LVH (e.g., increased QRS amplitude and “QRS widening”) and left-axis deviation. Predominantly, an HCM diagnosis is acquired through transthoracic echocardiography with Doppler, to assess the presence of LV outflow tract obstruction, mitral valve regurgitation, and diastolic dysfunction [72,73,74]. Utilization of MRI or angiogram can offer additional information if phenotypic features are difficult to determine and/or are inconclusive. Utilization of an MRI more adequately evaluates LV wall parameters (e.g., posterior, and anterior wall thickness), %EF, basal asymmetrical septal hypertrophy, apical HCM, and systolic, diastolic, and mitral valvular dysfunction. Lastly, LGE-MRI is a useful tool in the assessment of myocardial scar [73,74,75,76]. In the majority of HCM cases, the disease is autosomal dominant, although less common autosomal recessive and X-linked inheritance have been identified [77]. The first identified and most common gene mutation associated with HCM is *MYH7* [78,79]. Other gene-encoding proteins that are liable for HCM are *myosin binding protein-C* (*MYBPC3*) [80,81], and other sarcomeric proteins [82], which are also associated with DCM, such as *TNNT2* [83,84,85,86], *TNNI3* [87], *TNNC1* [88,89], *TPM1* [90], and *ACTC* [91]. Together, *MYH7* and *MYBPC3* genes account for nearly 50% of cases [77]. Similarly, to DCM, the majority of the HCM mutations are sarcomeric components or are associated with the contractile apparatus, making HCM and DCM distinguishable by their unique dysfunctional myocardial contractility [92]. Although troponin mutations have been associated with increased myofilament Ca^2+^ sensitivity [88], genetic penetrance also dictates phenotype severity (i.e., *TPM1* causes mild HCM) [93].

## 3. The Deleterious Impact of Exercise in ACM and Its Translational Potential in Murine Models

In complete contrast to decades of research on the beneficial impact of exercise on the cardiovascular system, patients with ACM are particularly at risk of increased arrhythmia, disease progression to HF, and SCD in response to exercise [94,95]. Prior studies in patients with ACM showed a strong correlation between exercise intensity and duration with worsening clinical phenotypes [96]. However, these bodies of work utilized retrospective clinical data in conjunction with self-reported exercise histories [94,95,96]. Although these clinical findings are a gold standard for understanding exercise influence on ACM phenotypes, they are merely correlative conjectures. 

Forced and high-intensity exercises can trigger pathological progression in ACM, not only due to exacerbated hemodynamic load under excessive physical activity but also as a consequence of psychological stress observed in ACM mice [97], which imparts additional concerns for research outcomes. A relevant consideration in patients diagnosed with ACM is the anxiety, psychosocial adjustment, and depression that contribute to both psychological burden and disease progression [98]. Ultimately, it is key to correlate both high-intensity exercise and psychological stress to facets of ACM disease phenotype worsening. Studies using animal models of ACM directly probed this causation, but the results involved forced exercise protocols (e.g., treadmill [99] and swimming [100,101]). In previous work, the unbridled physical effort (i.e., swimming) in *Dsg2*-mutant mice (*Dsg2*^mut/mut^) led to increased myocardial reactive oxygen species (ROS) generation and nuclear accumulation of apoptosis-inducing factor (AIF), prompting DNA fragmentation and cell death [102], demonstrating forced exercise and psychosocial stress are causal factors contributing to cardiac dysfunction and SCD in *Dsg2*^mut/mut^ mice [97,100,102]. Furthermore, Agrimi et al. demonstrated increased perceived psychosocial stress levels in ACM patients correlated with the extent of arrhythmias (e.g., premature ventricular contractions [PVCs] and supraventricular ectopics [SVEs]) and cardiac dysfunction (e.g., reduced right ventricular fractional area change [RVFAC]) [97]. In addition, other studies have demonstrated that stressful forced exercise contributes to HF progression and adverse cardiac function in mice [103,104].

A study on the evolution of ACM phenotypes published in 2020 demonstrates the effect of a 60-min treadmill exercise, which reversed cardiopathic transcripts levels and partially restored dysregulated genes in cardiac-restricted *Dsp* (*Myh6-Cre:Dsp*^W/F^) mice [105]. However, in other studies, endurance-trained mice with *Pkp2* mutations exhibited attenuation of ACM disease progression with worsening cardiac function and arrhythmias [101]. A *Dsp* mutation demonstrated the same influence under high-intensity exercise (i.e., treadmill until exhaustion), suggesting that exercise exacerbates ACM phenotype progression [106]. On the other hand, low-intensity exercise did not increase the risk of SCD in *Pkp2* mutant mice and indicated beneficial results [107]. We summarized studies using ACM murine models that were exposed to physical exercise (e.g., swimming or treadmill) and compared outcomes and molecular findings (Table 1).

High-intensity exercise impacts not only ACM progression but also other cardiovascular diseases considered that present with a high risk of SCD and fatal arrhythmias [108]. In studies of trained HCM mice, treadmill running demonstrated an overall improved response to the enhanced hemodynamic load, while forced swimming protocols revealed a concerning increase in fibrotic levels [109]. Similar results were observed by Konhilas et al. with HCM mice harboring mutant *Myhc* [110]. Voluntary exercise performed with DCM mice carrying mutant-*Tnnt2* indicated a beneficial cardiac function at younger ages, and prolonged survival [111]. Moreover, these same observations were demonstrated in α*Myhc* mice by Deluxe et al. where voluntarily exercised mice exhibited increased mitochondrial biogenesis via increased mitochondrial aconitase, voltage-dependent anion-selective channel-1 (VDAC1), and peroxisome proliferator-activated receptor-gamma coactivator-1α (PGC1α) levels, indicating slowed disease progression [112]. These exercise protocols indicate different outcomes depending on the cardiovascular disease model. We propound sarcomeric mutations show an overall improvement in cardiopathic phenotypes following exercise. Conversely, ACM, which has unique mutations associated with the ICD, demonstrates adverse differential outcomes dependent upon the exercise protocol. Thus, the assessment of voluntary exercise in animal models is fundamental to eliminating potential confounding variables, such as errors in ACM patients’ recollection of exercise or psychosocial stress reports and/or forced exercise protocols in ACM mice as seen in contemporaneous studies (Figure 1A–C). Voluntary exercise methods would likely provide more sound and translatable results [104,112,113]. Although it is understandable that it is difficult to encourage exercise in lazy mice with non-forced training protocols (Figure 1D), forced and stressful exercise has the potential for non-exercise mediated death (e.g., drowning), increased arrhythmic burdens (i.e., heart-brain axis) such as %PVCs/SVEs, and cardiac function (e.g., reduced %RVFAC). Current literature lacks studies that truly show causation with exercise and clinical phenotypes/SCD, and thus non-stressful protocols that can correlate distance run, time spent on the wheel, and speed/intensity with ACM phenotypes is of monumental importance.

On this note, high-intensity exercise is associated with pathological progression and worsening of disease phenotypes, exacerbated hemodynamic load, and psychological stress in ACM mice. Although low to moderate intensity exercise can slow disease progression, decrease SCD risk, delay cardiac remodeling, and increase mitochondrial biogenesis in DCM and HCM mice. In addition, forced exercise can induce cardiac dysfunction, increase myocardial remodeling, and elevate cardiac fibrosis in ACM, DCM, and HCM mice. Moreover, cardiac structural alterations are common in athletes of endurance/competitive sports (e.g., marathon, triathlon, cycling) with increased RV afterload during exercise [114,115]. On the other hand, voluntary exercise shows beneficial effects with ameliorated Ca^2+^ handling, phosphorylation levels of key contractile proteins, and improved cardiac function in ACM mice [107].

**Table 1 ijms-23-08753-t001:** Summary of contemporaneous studies on exercised ACM murine models and their respective molecular findings.

Gene Variant	Animal Model	Exercise Apparatus	Purpose of Study	Molecular Findings	Ref.
*Dsg2*	Homozygous *Dsg2*-mutant mice (*Dsg2*^mut/mut^)	Swimming	GSK3β-regulation	Reduced ICD signal for JUP, Cx43, and SAP97; all were normalized by SB216763 (GSK3β inhibitor). SB216763 treatment increased Cx43 protein levels. Abnormal GSK3β localization, and SB216763 normalizes it. *Dsg2*^mut/mut^ mice with constitutively active GSK3β demonstrated increased myocardial fibrosis and cardiac dysfunction.	[100]
	*Dsg2*^mut/mut^ mice	Swimming	Mitochondrial-mediated cell death	Calpain-1 (CAPN1) activation accounts for myocyte necrosis in exercised *Dsg2*^mut/mut^ mice. Mitochondrial dysfunction precedes Ca^2+^-mediated, CAPN1-induced necrosis. CAPN1 activation leads to AIF truncation, oxidation, and localization to myocyte nucleus in *Dsg2*^mut/mut^ mice and patients with ACM. Targeting cyclophilin-A prevents AIF nuclear import and reduces cell death in ACM myocytes.	[102]
*Pkp2*	Transgenic human-*PKP2* mice	Treadmill	Molecular defects associated with disease phenotype	Cardiac remodeling and reduction of Cx43 and Nav1.5 levels.	[99]
	AAV-R735X human-*PKP2* mice	Swimming	Impact of exercise on ACM cardiac manifestations	Cx43 delocalization at ICD.	[101]
	*Pkp2*^mut/+^ mice	Treadmill	Influence of exercise, pressure overload, and inflammation	Deficits in Ca^2+^-handling related proteins (CaV1.2, SERCA2a, AnkB, and Casq2). Exercise increased RV lateral Cx43 expression.	[116]
*Jup*	*Jup*^mut/+^ mice	Swimming	ACM development and desmosomal protein expression	RV contractile and EP function is altered without effects on Cx43 expression.	[117]
	*Jup*^mut/+^ mice	Swimming	Load-reducing therapy	Load-reducing therapy restores Cx43 phosphorylation levels.	[118]
*Dsp*	*Dsp*^mut/mut^ mice	Treadmill	Impact of loss-of-function *Dsp* mutation	Connexin-40 expression reduced in cardiomyocytes of the ventricular conduction system. Reduced total and phosphorylated Cx43 levels.	[106]
	Transgenic human-*DSP* mice.	Treadmill	Effects of endurance exercise on *DSP*-R2834H mutation	Focal fat in RVs and cytoplasmic DSP, JUP and Cx43 aggregates. Increased pGSK3β (Ser9) and p-AKT1 (Ser473) levels, at rest. Decreased nuclear GSK3β and AKT1 localization; reduced p-GSK3β, p-AKT1, and p-AKT1 (Ser308); loss of nuclear JUP after exercise. Exercise accelerates ACM pathogenesis and is associated with perturbed AKT1/GSK3β signaling.	[119]
	Myocyte-specific *Dsp* haplo-insufficient mice	Treadmill	Effects of treadmill exercise on cardiac phenotype.	Exercise restored two-thirds (n = 492/781, 63%) of transcript levels from differentially expressed genes, including epithelial–to-mesenchymal transition, inflammatory, and canonical WNT pathway genes.	[105]
*Dsc2*	*Dsc2*^mut/+^ and *Dsc2*^mut/mut^ mice	Treadmill	Effect of *Dsc2*-G790del mutation in ACM	No changes in DSC2, DSG2, JUP, PKP2, DSP, and Cx43 levels and localization between groups.	[120]

It is evident that it would be unethical to test, in real-time, the impact of exercise on arrhythmic burden and mortality in ACM patients. Thus, in short, a trial in ACM patients is impossible, and forced-exercise studies conducted in ACM animal models are contraindicated by stress-induced catecholamine/corticosterone release (i.e., heart-brain axis). More importantly, it is fundamental that researchers develop and conduct exercise protocols that properly bring out the disease phenotype without psychologically forcing it out. Therefore, voluntary exercise that tracks distance run, duration spent on wheels, and intensity (i.e., speed) should be used more frequently. The new Starr Life Science In-Cage Running Wheels (Figure 1E) is an experimental apparatus that can provide such exercise data that can then be correlated with cardiac functional parameters.

## 4. Correlation of Connexin-43 Expression and Exercise Training

In contrast to HCM and DCM, ACM is a disorder of the cardiomyocyte ICD, and it is associated with mutations in genes that encode proteins of the desmosomal complex, such as DSP, PKP2, JUP, DSC2, and DSG2 [11,17]. The desmosomal complex, along with AJs are responsible for connecting cardiomyocytes, therefore maintaining cell-cell electrical conduction and mechanical integrity within the cardiac tissue [13,121,122]. As such, mutations in ICD proteins promote disruption in cellular adhesion and electrical communication, ultimately, leading to cardiomyocyte cell death. Cardiomyocyte loss is a fundamental pathological mechanism for triggering the infiltration of immune cells and innate inflammation of the myocardial tissue, itself. The considerably damaged myocardium follows an abnormal fibrofatty deposition, acting as a flawed reparative response [123].

In addition to cell-cell adhesion, reduced expression of junction proteins is associated with impairment of electrical conduction and worsening of arrhythmias [124]. A specialized gap junction protein, Cx43, is critical for electrical propagation between cells, promoting appropriate cell-to-cell excitability in the heart [11,125]. Prior studies in ACM patients observed diminished signals of Cx43 at ICDs, indicating gap junction remodeling as an additional pathological feature of ACM [11,106]. In a study involving cardiac samples from DCM patients who died of SCD, the Cx43 signal was reduced in immunohistopathological samples, potentially correlating HF and arrhythmias [126]. A study involving LVH showed a compensatory increase of Cx43 in the initial stages of hypertrophy, with reduced Cx43 levels in chronically diseased hearts, which is potentially related to pressure overload remodeling [127]. Similar results were observed in LVH, where Cx43 levels increased due to phosphorylated isoforms (pCx43) in intermediate stages and decreased pCx43 levels at late stages of the disease [128,129]. Prior murine studies in the treatment of cardiac dysfunction targeting Cx43 demonstrated reduced disease progression via enhancing Cx43 expression in Duchenne muscular dystrophy mice [130]. Additionally, a study investigated the therapeutic efficacy of trans-endocardial injection of muscle-derived stem cells overexpressing the *Cx43* gene in human subjects with advanced HF, initial results showed promise in the potential of Cx43 as a therapeutic method for treating cardiac arrhythmias with increased myocardial viability, better exercise capacity, and potential for reverse remodeling [131]. Cx43 has also been shown to regulate sodium channel expression, thus reduced Cx43 leads to disturbances in sodium and potassium currents and mechanical coupling in cardiomyocytes [132]. 

Electrical conduction in ACM hearts is severely aggravated by the replacement of fibrofatty tissue at areas of cardiomyocyte loss, where progressive collagen deposition and myocardial scars act as an electrical barrier for impulse propagation [12,133]. Improving cardiac function by enhancing cell-to-cell communication is one effective strategy to ameliorate the cardiomyopathic phenotypes not only in ACM but also in HCM and DCM. Therefore, Cx43 may be a new key therapeutic in the treatment of arrhythmias and cardiac conduction disturbances [106]. Recent studies demonstrated increased Cx43 levels following moderate exercise, ameliorating overall myocardial dysfunction in mice with diabetic cardiomyopathy [134]. The same was observed in HCM studies with moderate-intensity exercise [135]; interestingly, however, high-intensity exercise resulted in remarkably decreased Cx43 levels [136,137]. Furthermore, repeated exhaustive exercise led to reduced cardiac conduction and ischemic injury, in conjunction with diminished Cx43 expression [138]. These results indicate that high-intensity, endurance exercise contributes to both cardiac dysfunction and myocardial injury, whereas moderate exercise benefits impulse conduction at a molecular level by increasing Cx43 expression. This is particularly relevant given that exercise intensity and duration correlate with worse cardiac phenotypes and survival outcomes in ACM patients, who already have reduced Cx43 levels. 

We would also like to bring up an excellent point made by Zorzi et al.: “It is difficult to infer from the majority of the studies whether there is a dose-response correlation between exercise and adverse outcome (i.e., ideally ACM patients should not exercise at all) or if there is a threshold of exercise type, intensity and/or duration that can be considered safe. However, the investigations that specifically addressed the outcome of ACM patients engaged in low-to-moderate intensity exercise found that their outcome was similar to that of sedentary patients. These findings suggest that not all exercise is the same and that patients with ACM may be able to derive the benefits of leisure time exercise without excess risk” [113].

In short, ACM patients have always been advised to avoid moderate-to-intense exercise to reduce arrhythmic burden, cardiac remodeling, and SCD, yet the sad irony is—the avoidance of low-to-moderate intensity exercise also eliminates the opportunity to increase Cx43 expression and improve cardiac function. Future studies are warranted to drive Cx43 expression in ACM patients. 

## 5. Therapeutics—Why Are We Unable to Advance beyond Antiarrhythmics in ACM?

ACM patients are at high risk of SCD. Regardless of the method of choice for ACM therapy, the ultimate goal is to reduce mortality. Over the years, clinicopathologic studies and reviews have thoroughly described lifestyle changes and clinical management options for ameliorating patients’ symptoms and improving their QOL. Endurance exercise contributes to the development and progression of disease phenotypes, in addition to inducing arrhythmias [113]. Contemporaneously, a study involving 129 ARVC patients with ICDs concluded that (i) “restriction of exercise should be recommended to ARVC patients regardless of mutation status and original indication for ICD, although patients who are gene-elusive and those with primary-prevention ICDs may particularly benefit”; and (ii) “exercise reduction is unlikely to reduce arrhythmia sufficiently in high-risk patients to alter decision-making regarding ICD implantation” [139]. More clinical trials on potentially safe exercise for ACM patients with ICDs are needed to further understand the correlation and causation between exercise and ACM.

Pharmacological treatment options for patients living with ACM are most often directed at preventing fatal ventricular arrhythmias (VAs), managing HF onset and progression, and aborting SCD/A. The latter is most preventable via an ICD [140]. Among the most vulnerable are ACM patients with a history of non-sustained (NSVT) and sustained VT, number of PVCs in 24 h via Holter monitor, prior cardiac syncope, number of ECG leads showing T-wave inversions, and RV dysfunction [140]. Numerous antiarrhythmics have been or are widely used in the treatment strategy of ACM patients, such as beta-blockers, sodium channel blockers, calcium channel blockers, amiodarone, and verapamil [141]. The most effective therapies are currently amiodarone, sotalol, beta-blockers, or some form of combination therapy (e.g., amiodarone/beta-blockers and/or flecainide/sotalol). Thus, antiarrhythmics are the most frequently used therapeutics in the management of ACM. Moreover, the advancement of ICDs since their first inception has allowed them to become truly life-saving devices in ACM and numerous forms of heart disease. 

For over two decades, the renin-angiotensin-aldosterone system has been seen as a therapeutic target for ACM patients with LV failure or HF [142]. Many studies have provided experimental and clinical evidence on the efficacy of ACEIs and/or ARBs [142,143,144]. More recently, a detailed retrospective observational cohort study including 311 patients diagnosed with ACM investigated whether ACEIs and/or ARBs blockers slow ACM progression and reduce the occurrence of VAs [145]. This was the first single-center observational study to provide insights into the effects of ACEIs and/or ARBs in patients with ACM during long-term follow-up. Although ACEIs and/or ARBs are not considered part of the classical armamentarium of ACM therapeutics, the authors found that patients with ACM treated with ACEIs and/or ARBs displayed a reduced risk of life-threatening VAs. In conclusion, ACEIs and/or ARBs might provide an additional benefit to those suffering from this devastating disease. 

However, this review would be remiss in not pointing out one grossly understudied, unexplored therapeutic arm—preventing fibrofatty remodeling. Given the importance of understanding the underlying mechanisms in which myocytes undergo intrinsic inflammation via NFĸB-mediated cytokine and chemokine expression and extrinsic inflammation via infiltrating T-cells and macrophages [4], followed by cell death then replacement by fibro-fatty tissue, considerable effort should be directed toward investigating new therapies focused on these pathophysiological hallmarks. Specifically, in those myocardial areas where inflammation and fat/fibrosis are not present, can we revitalize healthy heart cells while preventing the production of deleterious lipid and collagen deposition? Eventually providing precision medicine to ACM patients, such as dual therapeutics to manage arrhythmic burden and prevent inflammation-induced fibrotic remodeling. 

## 6. Conclusions

ACM is a unique disease with particular features that present as cardiac remodeling and fibrotic replacement of the myocardium. Currently, treatments are commonly focused on arrhythmia reduction and cardiac dysfunction, as well as improving the QOL of patients. However, recent research efforts have shown an alternative therapeutic target—Cx43—that appears to prevent pathological and functional hallmarks of ACM. This protein (i.e., Cx43) improves cellular adherence and prevents cardiomyocyte loss. By preventing cardiomyocyte cell death, both inflammation and fibrosis could be suppressed, thus resulting in an improvement in overall cardiac function. Studies show that increased Cx43 expression can be achieved by low-to-moderate intensity exercise, which ACM patients are advised against. In this review, Cx43 is highlighted as a potential therapeutic target for treating cardiac arrhythmias and increasing myocardial viability in ACM patients. Additionally, we offer hope in safe, alternative approaches to exercise for patients. Although there is, indeed, cause for concern to increase RV hemodynamic load as well as elevated heart rate in response to physical efforts, recent research has shown significant benefits for low-to-moderate exercise. This is especially true in the accompaniment of load-reducing and/or ARBs/ACEIs; the latter showed significant promise in small cohort of ACM patients. Therefore, we conclude with a call for researchers and physician-scientists to investigate this possibility—preventing the development of inflammation and fibrofatty infiltration. In particular, by seeking and enrolling asymptomatic gene carriers and/or low-risk ACM patients in non-antiarrhythmic trials; alternatively, ACM patients with an ICD and for which antiarrhythmics are unable to reduce arrhythmic burden.

## Figures and Tables

**Figure 1 ijms-23-08753-f001:**
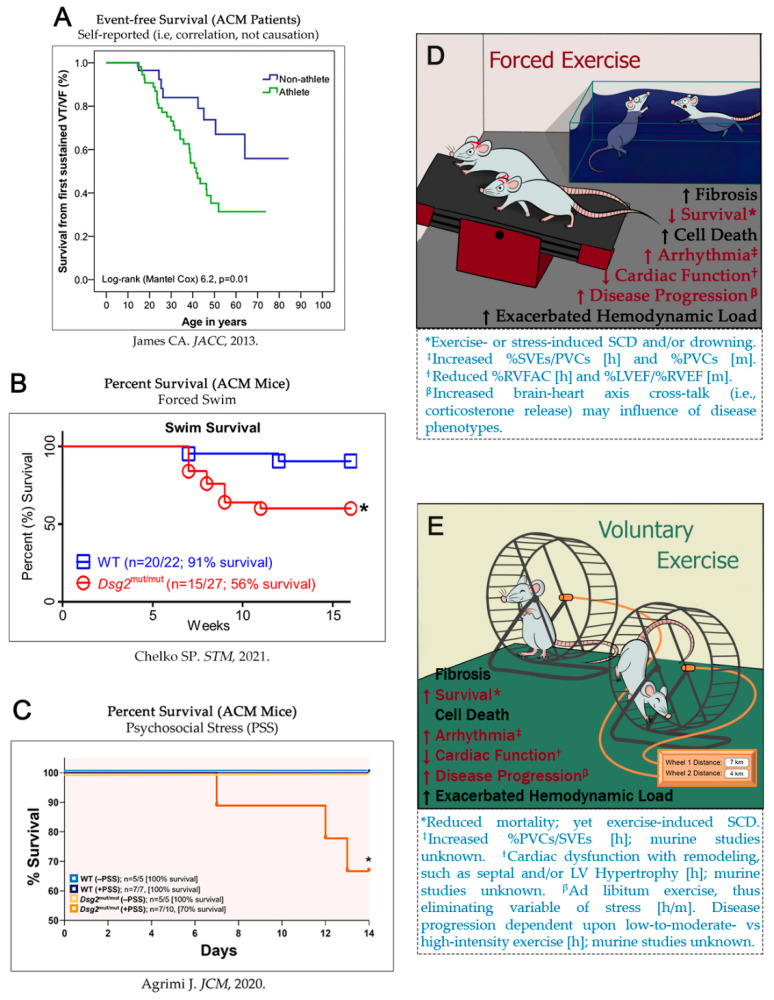
Survival analyses in ACM subjects. (**A**) Event-free survival from VT/VF in ACM patients. (**B**) Percent survival in ACM mice subjected to forced swim or (**C**) psychosocial stress. Graphical abstract demonstrating mice under stress whilst performing forced exercises (**D**), during voluntary exercise (**E**) demonstrates no psychological and/or psychosocial stress in mice. For (**D**,**E**), prior research outcomes *, ^‡^, ^†^, ^β^ correspond with the text immediately below the image. [h], human; [m] murine studies. For (**A**–**C**), images are reprinted with permission from (**A)** [94] (2103, Elsevier), (**B)** [102] (2021, Chelko S.P.) and (**C**) [97] (2020, Chelko S.P.).

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
