# Peer review of "Arrhythmogenic Cardiomyopathy: Exercise Pitfalls, Role of Connexin-43, and Moving beyond Antiarrhythmics"

_ijms, 2022, doi:10.3390/ijms23158753_

Round 1
Reviewer 1 Report
The paper presents an appealing study on muscular dysfunction.
The manuscript is well written and the results could be of interest for research and potentially also for practice.
A major strength represents the thorough physiological perspective applied.
I have to admit that I have no experience with research on mice.
A more detailed investigation how the present findings might apply to human patients could strengthen the paper.
Potentially, some examples from everyday life could also be discussed in more depth.
Author Response
We have additionally uploaded a PDF of our Response.
RESPONSE TO REVIEWERS
We sincerely thank the reviewers for providing challenging critiques to improve the quality of our review, which we have significantly enhanced as a result. This all serves our effort to advance scientific understanding of Arrhythmogenic Cardiomyopathy (ACM), the deleterious role of prolonged physical efforts in ACM patients, evaluating the emergence of refined exercise protocols for translational animal studies, and the need for precision medicine therapeutics.
Our responses to each Reviewer are given, point-by-point below. We have also provided any significant changes within the manuscript in blue type with corresponding line references and pagination below. We hope that this revised version meets both the expectations and the high publication standards of IJMS.
Reviewer #1: “The paper presents an appealing study on muscular dysfunction. The manuscript is well written and the results could be of interest for research and potentially also for practice. A major strength represents the thorough physiological perspective applied. I have to admit that I have no experience with research on mice.”
- “A more detailed investigation how the present findings might apply to human patients could strengthen the paper.”
Response: This is an important point to be made, as such we have added the following section – “6. Conclusion” – to the end of the manuscript (pg.11; lines 424 – 445):
“ACM is a unique disease with particular features that present as cardiac remodeling and fibrotic replacement of the myocardium. Currently, treatments are commonly focused on arrhythmia reduction and cardiac dysfunction, as well as improving the QOL of patients. However, recent research efforts have shown an alternative therapeutic target – Cx43 – that appears to prevent pathological and functional hallmarks of ACM. This protein (i.e., Cx43) improves cellular adherence and prevents cardiomyocyte loss. By prevention of cardiomyocyte cell death, both inflammation and fibrosis could be suppressed, and thus an improvement in overall cardiac function. Studies show that increased Cx43 ex-pression can be achieved by low-to-moderate intensity exercise, for which ACM patients are advised against. In this review, Cx43 is pointed as a potential therapeutic target for treating cardiac arrhythmias and increasing myocardial viability in ACM patients. Additionally, we offer hope in safe, alternative approaches to exercise for patients. While there is, indeed, cause for concern to increase RV hemodynamic load as well as elevated heart rate in response to physical efforts, recent research has shown significant benefits for low-to-moderate exercise. Especially in the accompaniment of load-reducing and/or ARBs/ACEIs. The latter of which showed significant promise in small cohort of ACM patients. Therefore, we conclude with a call for researchers and physician scientists to investigate this possibility – preventing the development of inflammation and fibrofatty infiltration. Particularly, by seeking and enrolling asymptomatic gene-carriers and/or low-risk ACM patients in non-antiarrhythmic trials; alternatively, ACM patients with an ICD and for which anti-arrhythmics are unable to reduce arrhythmic burden.”
- “Potentially, some examples from everyday life could also be discussed in more depth.”
Response: We thank the reviewer for highlighting this aspect of everyday life. As there are, sadly, a plethora of life-style changes ACM patients will encounter in their lives. We have added the following paragraph to section “2. The Cardiomyopathies” (pg. 3; lines 127 – 137):
“In addition to the cardiac phenotypes ACM patients develop, there are, sadly, QOL changes ACM patients will encounter and for some, must implement in everyday life. For example, patients whom recently had VT storm or an ICD discharge are restricted from driving. While driving restrictions are dependent upon the state in which the patient re-sides, driving restrictions can range from 3-12 months. Although we will discuss exercise guidelines below, even a simple day at the beach or lounging at the pool can be a life-altering event. Syncope, VT storm, or an ICD shock may limit a patient’s independence, as a “buddy system” is essential – even at a depth of 2 inches of water – for fear of drowning. Furthermore, ACM patients often confront the burdens of psychological stress, such as ICD implantation, ICD shocks (e.g., appropriate or inappropriate discharges) and starting a family – as trepidation of passing down a pathogenic variant is possible.”

Reviewer 2 Report
The review overviews the intriguing impact of exercise in the pathogenesis and treatment of clinical and translational arrhythmogenic cardiomyopathy (ACM) causing sudden cardiac death. Mechanistically, the authors focus only on the cardiac connexin-43 mechanism of ACM’s pathogenesis. However, the review title does not represent the content, and the review is not focused and not well organized.
The Abstract and Introduction should specify if the authors refer to ACM as the pathology of the right and left or both ventricles or to the pathology affecting only right ventricle - arrhythmogenic right ventricular cardiomyopathy (ARVC).
ACM and ACM progressed to heart failure should be introduced and distinguished in the review.
The Introduction is not entirely aligning with the review sub-section and must be re-written. For instance, the role of inflammation in progressive fibrofatty replacement of the myocardium associated with ACM is introduced in the Introduction. Still, it has not been undressed in any review sub-section.
The statements on lines 40-46 must be supported by references.
Section 2: The Cardiomyopathies
The subsections under this section should be better structured and logically connected. Overall this section should be focused on ACM with DCM and HCM as contributors to ACM phenotype since this pathology affects the right ventricle, left ventricle, or both.
DCM and HCM subsections must be significantly rewritten. Below you will see just a few concerns, among many others:
Lines 62-64: These two sentences are not logically connected: “DCM can manifest as an ischemic or non-ischemic disease, where ischemic DCM manifests following extensive myocardial necrosis [6]. Conversely, non-ischemic DCM represents most clinical cases, and around 35% of these cases are familial inherited, while the other 65% are acquired throughout the patient’s life.”
Plus, the ischemic DCM is associated with not only myocardial necrosis but also myocardial apoptosis. The myocardial cell death is mainly related to the acute stage of MI rather than DCM.
Lines 66-68: heart failure is not a DCM symptom but the final stage of DCM progression.
The biomarker should be clarified as plasma biomarkers or cardiac transcript level/biomarker.
Lines 68-94 - related to ECG and DCM biomarkers – must be significantly rewritten for logical flow and contest. Thus, MYH7 is a marker of symptomatic heart failure rather than DCM.
Section 3: The Deleterious Impact of Exercise and Its Translational Potential in Murine 173 Models
It would be reasonable to incorporate subsections:
- High-intensity exercise vs. Moderate and low-intensity exercise;
- Forced vs. voluntary exercise.
The quality of Fig. 1 A, B, and C must be improved since the incorporated text is unreadable.
Figure 1 D should not only present images of the forced and voluntary exercise but also demonstrate their outcomes at ACM.
Table 2: Please structure the findings (bullet points).
Section 4: Correlation of Connexin-43 Expression and Exercise Training
The sub-section related to fibrofatty replacement is missed and required.
Line 344 – renin-angiotensin-aldosterone system rather than renin-angiotensin system.
The conclusions subsection is required for readers' benefit.
Author Response
We have additionally uploaded a PDF of our Response.
RESPONSE TO REVIEWERS
We sincerely thank the reviewers for providing challenging critiques to improve the quality of our review, which we have significantly enhanced as a result. This all serves our effort to advance scientific understanding of Arrhythmogenic Cardiomyopathy (ACM), the deleterious role of prolonged physical efforts in ACM patients, evaluating the emergence of refined exercise protocols for translational animal studies, and the need for precision medicine therapeutics.
Our responses to each Reviewer are given, point-by-point below. We have also provided any significant changes within the manuscript in blue type with corresponding line references and pagination below. We hope that this revised version meets both the expectations and the high publication standards of IJMS.
Reviewer #2: “The review overviews the intriguing impact of exercise in the pathogenesis and treatment of clinical and translational arrhythmogenic cardiomyopathy (ACM) causing sudden cardiac death. Mechanistically, the authors focus only on the cardiac connexin-43 mechanism of ACM’s pathogenesis. However, the review title does not represent the content, and the review is not focused and not well organized.”
Response: We are sorry to hear this, and we apologize for any confusion on the title. When first contacted by IJMS regarding a Review submission we suggested “From Desmosomal Disruption to Muscle Dysfunction.” We regret, not on our own accord, to see that this title is on the submission homepage. However, we found it more important to highlight the comparisons of ACM to HCM and DCM, emerging evidence of the deleterious impact of exercise in ACM, and the failure of the field to move beyond anti-arrhythmics. As such, we settled on the title “Arrhythmogenic Cardiomyopathy: Exercise Pitfalls and Moving Beyond Antiarrhythmics,” which was included in our Cover letter and our Original Manuscript submission. While we did not enter “From Desmosomal Disruption to Muscle Dysfunction” on the submission site, we sorely regret that we did not see this and change it accordingly. That said, we have changed our title to “Arrhythmogenic Cardiomyopathy: Exercise Pitfalls, Role of Connexin-43 and Moving Beyond Antiarrhythmics.” Additionally, we have re-organized the structure of the paper to adequately align with the subsections established in the Introduction.
- “The Abstract and Introduction should specify if the authors refer to ACM as the pathology of the right and left or both ventricles or to the pathology affecting only right ventricle - arrhythmogenic right ventricular cardiomyopathy (ARVC).”
Response: We thank the Reviewer for highlighting this point. The relevant terminology has been revised and now the text clearly indicates the specifics of the pathology. Specifically, we have edited the following text for clarification (pg. 2; lines 72 – 75):
“Arrhythmogenic Cardiomyopathy, commonly referred to as Arrhythmogenic Right Ventricular Cardiomyopathy (ARVC), is a myocardial disorder that can affect either the right (ARVC), the left (ARLV), or both ventricles; hence its recent and more inclusive nomenclature - ACM.”
- “ACM progression to heart failure should be introduced and distinguished in the review.”
Response: This is of great importance, as the field originally proposed that ACM patients do not develop HF or, at least, its rarity. Yet, recent work has provided additional studies to highlight the eventual HF progression these patients develop. The subsection has been revised, and the following information regarding HF in ACM has been added to the review paper (pg. 3; lines 105 – 109):
“Regardless of index presentation, LV dysfunction is a common phenotype at time of transplantation; where, the most frequent attestation for transplant is heart failure (HF) [22]. Although HF was originally considered to be rare in ACM [23], recent studies by Gilotra NA et al demonstrated that at least one symptom of HF was present in 49% (n=142/289) of ACM patients [24].”
- “The Introduction is not entirely aligning with the review sub-section and must be re-written. For instance, the role of inflammation in progressive fibrofatty replacement of the myocardium associated with ACM is introduced in the Introduction. Still, it has not been undressed in any review sub-section.”
Response: The introduction has been revised to better align with the flow of the manuscript’s subsections. More specifically, we have moved certain sections to follow their outline as described in the Introduction, also addressing the therapeutics subsection within the Introduction. We modified the review to now include both intrinsic (i.e., originating from the myocytes, themselves) and extrinsic (i.e., infiltrating immune cells) mechanisms of inflammation to the manuscript.
- “The statements on lines 40-46 must be supported by references.”
Response: The text is now properly cited. More specifically, references 3 – 9 have now been added.
Section 2: The Cardiomyopathies
- “The subsections under this section should be better structured and logically connected. Overall this section should be focused on ACM with DCM and HCM as contributors to ACM phenotype since this pathology affects the right ventricle, left ventricle, or both.”
Response: We thank the reviewer for their comments and this subsection has been revised substantially. That said, the authors would like to state that “…DCM and HCM [does not act] as contributors to ACM phenotype…”. ACM is an entirely different heart disease where over 60% of patient cases consist of pathogenic desmosomal variants not assigned to DCM and HCM. However, we have added the following statement (pg.3; lines 137 – 140):
“While not all of these QOL changes are applicable to all heart diseases, ACM presents with considerable phenotypic overlap with the other two cardiomyopathies: hypertrophic (HCM) and dilated cardiomyopathy (DCM).”
- “DCM and HCM subsections must be significantly rewritten. Below you will see just a few concerns, among many others: Lines 62-64: These two sentences are not logically connected: “DCM can manifest as an ischemic or non-ischemic disease, where ischemic DCM manifests following extensive myocardial necrosis [6]. Conversely, non-ischemic DCM represents most clinical cases, and around 35% of these cases are familial inherited, while the other 65% are acquired throughout the patient’s life.”
Response: We thank the Reviewer for their careful critique of this sentence, as it can imply 65% of non-ischemic DCM cases could very well be acquired via a myocardial infarction – thus, qualifying it as ischemic-DCM. We have listed the major causes of acquired DCM as well. This subsection have been revised and rewritten.
- “Plus, the ischemic DCM is associated with not only myocardial necrosis but also myocardial apoptosis. The myocardial cell death is mainly related to the acute stage of MI rather than DCM.”
Response: Again, we thank the Reviewer for their careful and grammatical overview of this section. Grammar and punctuation truly matter, thank you for catching this mistake. The sentence has been revised accordingly (see pg 3; lines 141 – 149).
- “Lines 66-68: heart failure is not a DCM symptom but the final stage of DCM progression.”
Response: The sentence has been revised accordingly and has been edited as follows (pg. 4; now lines 151 – 153):
“Although DCM patients present lower burden of comorbidities, HF – more specifically HF with reduced ejection fraction – is the most common cause of mortality in DCM patients [38].”
- “The biomarker should be clarified as plasma biomarkers or cardiac transcript level/biomarker.”
Response: The sentence has been revised accordingly. And now provides further clarification of its specimen origin, which is serum and not plasma (pg. 4; lines 164 – 166):
“An additional clinical presentation of DCM is elevated levels of troponin-T (TNNT), a serum biomarker indicative of cardiomyocyte degeneration [48].”
- “Lines 68-94 - related to ECG and DCM biomarkers – must be significantly rewritten for logical flow and contest. Thus, MYH7 is a marker of symptomatic heart failure rather than DCM.”
Response: The section was revised accordingly (see pg. 4; now lines 176 – 178).
Section 3: The Deleterious Impact of Exercise and Its Translational Potential in Murine Models.
- “It would be reasonable to incorporate subsections:
- High-intensity exercise vs. Moderate and low-intensity exercise;
- Forced vs. voluntary exercise.”
Response: These subsections have been revised, moved around and additional items addressed. Please refer to pages 5 and 6; lines 211 – 196).
- “The quality of 1 A, B, and C must be improved since the incorporated text is unreadable.”
Response: We apologize for this issue. We have embedded a higher quality figure, in addition to uploading a high-tiff figure to the submission website. Lastly, we have added an editable Figure 1 Legend to the end of the manuscript before the references, in case the Reviewers and/or Editorial board wants the authors to revise this legend.
- “Figure 1Dshould not only present images of the forced and voluntary exercise but also demonstrate their outcomes at ACM.”
Response: We concur with the Reviewer. Outcomes of forced and voluntary exercises are now shown on the figure in brief written form. Additionally, an extended detail of the outcomes can be found on Table 1.
- “Table 1:Please structure the findings (bullet points).”
Response: All the molecular findings are now listed as bullet points.
Section 4: Correlation of Connexin-43 Expression and Exercise Training
- “The sub-section related to fibrofatty replacement is missed and required.”
Response: We agree with the Reviewer that a, ”…sub-section related to fibrofatty replacement is missed…”. However, we kindly disagree with the critique that a section solely focused on fibrofatty replacement is “required.” This unique pathological hallmark of ACM would require a separate Review on the origins of AVRD (i.e., dysplasia) to now its understanding as a myopathy (i.e., progressive fat infiltration with muscle weakening) and the plethora of papers detailing its mechanistic etiology – which is still elusive. That said, this subsection has been revised accordingly, to add information regarding its deposition. Our primary goal was to compare and contrast ACM with DCM/HCM, highlight the influence of exercise, and the need for better therapeutics. Please refer to pg. 8, lines 316 – 319.
Section 5: Therapeutics - Why Can’t We Advance Beyond Antiarrhythmics
- “Line 344 – renin-angiotensin-aldosterone system rather than renin-angiotensin system.”
Response: The sentence has been revised accordingly.
Conclusion:
- “The conclusions subsection is required for readers' benefit.”
Response: We agree with the Reviewers’ suggestion and we regret we did not include this in the original submission. The benefit of adding a conclusion subsection is of monumental importance, as such it has been added to the manuscript as follows (pg. 11; lines 424 – 445):
“ACM is a unique disease with particular features that present as cardiac remodeling and fibrotic replacement of the myocardium. Currently, treatments are commonly focused on arrhythmia reduction and cardiac dysfunction, as well as improving the QOL of patients. However, recent research efforts have shown an alternative therapeutic target – Cx43 – that appears to prevent pathological and functional hallmarks of ACM. This protein (i.e., Cx43) improves cellular adherence and prevents cardiomyocyte loss. By prevention of cardiomyocyte cell death, both inflammation and fibrosis could be suppressed, and thus an improvement in overall cardiac function. Studies show that increased Cx43 ex-pression can be achieved by low-to-moderate intensity exercise, for which ACM patients are advised against. In this review, Cx43 is pointed as a potential therapeutic target for treating cardiac arrhythmias and increasing myocardial viability in ACM patients. Additionally, we offer hope in safe, alternative approaches to exercise for patients. While there is, indeed, cause for concern to increase RV hemodynamic load as well as elevated heart rate in response to physical efforts, recent research has shown significant benefits for low-to-moderate exercise. Especially in the accompaniment of load-reducing and/or ARBs/ACEIs. The latter of which showed significant promise in small cohort of ACM patients. Therefore, we conclude with a call for researchers and physician scientists to investigate this possibility – preventing the development of inflammation and fibrofatty infiltration. Particularly, by seeking and enrolling asymptomatic gene-carriers and/or low-risk ACM patients in non-antiarrhythmic trials; alternatively, ACM patients with an ICD and for which anti-arrhythmics are unable to reduce arrhythmic burden.”

Round 2
Reviewer 2 Report
The authors satisfactorily addressed most of my critiques and suggestions. Thank you.
Still, Figure 1 panels D and E require additional impute since it remains unclear whether exercises positively or negatively impact fibrosis, cell death, cardiac remodeling, and cardiac dysfunction.
Author Response
Response: We thank the Reviewer for highlighting this lack of clarity in our revised Figure. We’ve added additional text to the Figure Legend and directly below Fig. 1D, E. Specifically, changes previously made in bold Red Text in Fig. 1D, E now address outcomes from murine and human studies in blue font below. We thought a simple highlighting of these differences would suffice, but now realize the inadequacy of our visual explanation. We have now revised these changes by adding “↑ or ↓” next to the figure text in Fig. 1D to represent an increase or decrease in a functional or pathological parameter. Or did not include the “↑ or ↓” symbols at all (e.g., Fig 1E) if there has been no prior work demonstrating these effects in voluntary exercise in mice. We have added the following text to the figure and figure legend, respectively:
Figure 1D:
*Exercise- or stress-induced SCD and/or drowning. ‡Increased %SVEs/PVCs [h] and %PVCs [m]. †Reduced %RVFAC [h] and %LVEF/%RVEF [m] . βIncreased brain-heart axis cross-talk (i.e., corticosterone release) may influence disease phenotypes.
Figure 1E:
*Reduced mortality; yet exercise-induced SCD. ‡Increased %PVCs/SVEs [h]; murine studies unknown. †Cardiac dysfunction with remodeling, such as septal and/or LV Hypertrophy [h]; murine studies unknown. βAd libitum exercise, thus eliminating variable of stress [h/m]. Disease progression dependent upon low-to-moderate- vs high-intensity exercise [h]; murine studies unknown.
Figure Legend:
For (D, E), prior research outcomes *‡†β correspond with text immediately below image. [h], human; [m] murine studies.”
